# The bZIP Transcription Factor GmbZIP15 Negatively Regulates Salt- and Drought-Stress Responses in Soybean

**DOI:** 10.3390/ijms21207778

**Published:** 2020-10-21

**Authors:** Man Zhang, Yanhui Liu, Hanyang Cai, Mingliang Guo, Mengnan Chai, Zeyuan She, Li Ye, Yan Cheng, Bingrui Wang, Yuan Qin

**Affiliations:** 1Key Lab of Genetics, Breeding and Multiple Utilization of Crops, Ministry of Education, State Key Laboratory of Ecological Pest Control for Fujian and Taiwan Crops, Fujian Provincial Key Laboratory of Haixia Applied Plant Systems Biology, Center for Genomics and Biotechnology, College of Plant Protection, College of Life Sciences, College of Crop Science, Fujian Agriculture and Forestry University, Fuzhou 350002, China; zhangman3043@163.com (M.Z.); yanhuiliu520@gmail.com (Y.L.); caihanyang123@163.com (H.C.); gml604@163.com (M.G.); chaimengnan1@163.com (M.C.); 1180204044@fafu.edu.cn (L.Y.); chengyan1220@hotmail.com (Y.C.); 2State Key Laboratory for Conservation and Utilization of Subtropical Agro-Bioresources, Guangxi Key Lab of Sugarcane Biology, College of Agriculture, Guangxi University, Nanning 530004, China; szy0809@yeah.net; 3College of Plant Science & Technology, Huazhong Agricultural University, Wuhan 430070, China

**Keywords:** GmbZIP15, transcription factor, salt stress, drought stress, RNA-seq, soybean

## Abstract

Soybean (*Glycine max*), as an important oilseed crop, is constantly threatened by abiotic stress, including that caused by salinity and drought. bZIP transcription factors (TFs) are one of the largest TF families and have been shown to be associated with various environmental-stress tolerances among species; however, their function in abiotic-stress response in soybean remains poorly understood. Here, we characterized the roles of soybean transcription factor GmbZIP15 in response to abiotic stresses. The transcript level of *GmbZIP15* was suppressed under salt- and drought-stress conditions. Overexpression of *GmbZIP15* in soybean resulted in hypersensitivity to abiotic stress compared with wild-type (WT) plants, which was associated with lower transcript levels of stress-responsive genes involved in both abscisic acid (ABA)-dependent and ABA-independent pathways, defective stomatal aperture regulation, and reduced antioxidant enzyme activities. Furthermore, plants expressing a functional repressor form of *GmbZIP15* exhibited drought-stress resistance similar to WT. RNA-seq and qRT-PCR analyses revealed that *GmbZIP15* positively regulates *GmSAHH1* expression and negatively regulates *GmWRKY12* and *GmABF1* expression in response to abiotic stress. Overall, these data indicate that GmbZIP15 functions as a negative regulator in response to salt and drought stresses.

## 1. Introduction

As a result of their sessile nature, plants are subject to variable biotic and abiotic-stress conditions. As the most pertinent abiotic-stress conditions, drought and salinity threaten the growth and productivity of crops. Plants respond and adapt to these stress conditions by activating stress-related pathways, which comprise signal perception and transduction, regulation of gene expression, and biochemical and physiological responses [1]. Signaling pathways, including those involving various phytohormones, multiple secondary metabolism processes, and reactive oxygen species (ROS), are crucial for plant survival under environmental-stress conditions [2,3,4].

Abiotic stress usually leads to the generation of ROS such as hydrogen peroxide (H_2_O_2_) and superoxide (O^2−^); however, ROS overaccumulation is cytotoxic [5,6]. To control the level of ROS accumulation under stress conditions, plants have evolved a wide range of antioxidants to scavenge ROS and reinstate cellular redox homeostasis. Previous studies have found that ROS signaling is linked to abscisic acid (ABA), Ca^2+^ fluxes, and sugar sensing, and is likely to be involved in ABA-dependent signaling pathways activated under abiotic stress [5,7]. Overexpression of *GmSIN1* in soybean (*Glycine max*) promotes root growth and salt tolerance by enhancing cellular ABA and ROS contents [7]. In rice (*Oryza sativa*), OsCPK12 promotes salt-stress resistance, likely through repression of ROS production and/or the participation of the ABA signaling pathway [8].

ABA is one of the most important stress-related phytohormones and plays a pivotal role in signal transduction during abiotic-stress responses [8,9]. Cellular ABA levels in plants increase in response to abiotic stress, leading to the expression of stress-responsive genes, the regulation of metabolic processes, and the quenching of ROS, thereby maintaining plant cell homeostasis under stress conditions [10]. Recent studies have found that plants respond to abiotic-stress conditions mainly through ABA-dependent and ABA-independent signaling pathways, which are regulated by AREB/ABFs and DREB2A transcription factors (TFs), respectively [11]. In *Arabidopsis* under abiotic-stress conditions, ABA enacts responses primarily through four bZIP TFs, namely, ABF1, AREB1/ABF2, AREB2/ABF4, and ABF3, the expression of which is activated by SnRK2s [10,11,12,13,14]. Abiotic stress and ABA has also been shown to induce the expression of *AREB1*/*ABF2*, *AREB2*/*ABF4*, and *ABF3* [15], and an increased abundance of these AREB/ABFs in turn increases ABA sensitivity and abiotic-stress resistance [13,16]. *ABF1* has also been shown to function in drought-stress responses, despite its lower expression level compared to other abiotic-stress-induced AREB/ABFs, and thus *areb1areb2abf3abf1* plants display decreased drought resistance compared to *areb1areb2abf3* plants regarding primary root growth [10]. DREB2 proteins are members of the AP2/ERF family of plant-specific TFs and function in an ABA-independent manner [11]. Among the DREB2 genes in *Arabidopsis*, DREB2A is largely induced by drought, salinity, and cold stress [17]. Three DREB homologues, namely, *GmDREBa*, *GmDREBb*, and *GmDREBc*, have been identified in the soybean genome and the transcript levels of *GmDREBa* and *GmDREBb* in the leaves of soybean seedlings were shown to increase following salt-, drought-, and cold-stress treatment [18].

Transcription factors are considered to be the most important regulators of gene expression. Several groups of TFs, such as DREB, NAC, MYB, WRKY, and bZIP, are responsible for abiotic-stress responses [19,20,21,22,23]. The bZIP TFs, which represent one of the largest plant TF families, can be divided into different subfamilies depending on the bZIP domain [24]. Plant bZIP TFs play crucial regulatory roles in multiple abiotic-stress resistances [12,25,26]. In *Arabidopsis*, *ABI5* expression is regulated by *ABF3*, which may contribute to salt-stress tolerance [14]. In rice, OsABF1 improves drought tolerance by activating the transcription of *COR413-TM1* [12]. Overexpression of the sweet potato (*Ipomoea batatas*) TF *IbABF4* in *Arabidopsis* resulted in increased ABA sensitivity as well as enhanced drought and salt-stress tolerance [27].

Soybean is one of the most important crops and is widely cultivated worldwide because of its nutritive value. However, soybean growth is threatened by many abiotic-stress factors such as salinity, drought, and extreme temperature. In a previous study, 160 bZIP family members were identified from the soybean genome and were divided into 12 subgroups [24]. Among these, many family members have been characterized to play roles in abiotic-stress responses, including *GmbZIP132*, *GmbZIP110*, *GmbZIP44*, *GmbZIP62,* and *GmbZIP78* [28,29,30]. However, the function of GmbZIP15, the only member of subfamily K, in response to abiotic stress remains poorly understood. In this study, biochemical and physiological analyses were performed to reveal the regulatory roles of GmbZIP15 in abiotic-stress responses.

## 2. Results

### 2.1. GmbZIP15 Expression Pattern in Response to Abiotic-Stress Conditions

We previously identified 160 bZIP genes from soybean and characterized their expression in response to abiotic stress [24]. Among these genes, the expression of *GmbZIP15* was suppressed by drought and flooding stress and it was therefore selected for further investigation. To validate the response of *GmbZIP15* to abiotic-stress conditions, we generated soybean plants overexpressing GmbZIP15 (*OX-GmbZIP15*), and two lines (OE-8, OE-16) with higher expression level were selected for further research (Appendix A). The qRT-PCR analysis was used to determine *GmbZIP15* expression patterns in 2-week-old wild-type (WT) and two overexpression lines under salt and drought treatments. Under normal conditions, *GmbZIP15* expression level in *OX-GmbZIP15* plants was clearly higher than that in WT; however, *GmbZIP15* expression in *OX-GmbZIP15* plants treated with NaCl and mannitol sharply decreased, but was higher than WT plants by 12 h and was nearly undetectable by 24 h post-treatment (Appendix A). In addition, through the GUS staining of 1-week-old *pGmbZIP15:GUS* transgenic *Arabidopsis* seedlings grown on media supplemented with 100 mM NaCl or 200 mM mannitol, *GmbZIP15* promoter activity was observed to be significantly decreased in cotyledons and true leaves under NaCl and mannitol conditions compared to normal conditions (Appendix A), which was consistent with the results of qRT-PCR (Appendix A). These results suggest that the expression of *GmbZIP15* is suppressed by abiotic stress.

### 2.2. GmbZIP15 Negatively Regulates Salt and Drought Tolerance in Soybean

To investigate the role of GmbZIP15 in plant response to salt stress, transgenic soybean plants carrying functional repression of *GmbZIP15* (*35S:GmbZIP15-SRDX*) were obtained and two lines (SRDX-15, SRDX-21) with higher expression levels were selected for further research (Appendix A). WT, *OX-GmbZIP15* and *35S:GmbZIP15-SRDX* seedlings were treated with 200 mM NaCl. Following 2 weeks of salt treatment, WT and *35S:GmbZIP15-SRDX* soybean seedlings exhibited a comparable degree of leaf shedding, whereas *OX-GmbZIP15* plants displayed a severe, almost lethal wilt phenotype (Figure 1A). These results suggest that overexpression of *GmbZIP15* in soybean causes sensitivity to salt stress.

To test the function of GmbZIP15 in plant drought responses, WT, *OX-GmbZIP15* and *35S:GmbZIP15-SRDX* seedlings were withheld water for 2 weeks. Compared to WT and *35S:GmbZIP15-SRDX* seedlings, *OX-GmbZIP15* seedlings were severely wilted and almost all leaves displayed a considerable dehydration phenotype (Figure 2A). To test whether the dehydration phenotype could be rescued, we rewatered the drought-treated plants for 3 days. Although there was slight shedding of the old leaves of WT and *35S:GmbZIP15-SRDX* seedlings after rewatering, there was vigorous new leaf growth, suggesting that WT and *35S:GmbZIP15-SRDX* plants can recover from such a dehydration phenotype. In contrast, *OX-GmbZIP15* seedlings did not display growth recovery after rewatering. Combined, these results further support that GmbZIP15 acts as a potential negative regulator of abiotic-stress response.

In addition, the transcription levels of drought-/salt-tolerance marker genes, including *GmDREBb*, *GmMYB118,* and *GmWRKY28*, were evaluated by qRT-PCR in WT and *OX-GmbZIP15* plants. Under mock conditions, the expression levels of each marker were considerably higher in *OX-GmbZIP15* plants compared to WT plants (Appendix A). However, under salt stress, the expression levels of *GmDREBb*, *GmMYB118,* and *GmWRKY28* dramatically decreased in *OX-GmbZIP15* plants compared with that in WT plants (Appendix A). Similarly, under drought stress, the expression levels of *GmDREBb*, *GmMYB118,* and *GmWRKY28* genes in *OX-GmbZIP15* plants also decreased significantly compared with that in WT plants (Appendix A). These results indicate that *GmbZIP15*-overexpressing soybean plants are hypersensitive to abiotic stress.

### 2.3. GmbZIP15 Depresses the ROS Scavenging Ability of Soybean

Abiotic stress can lead to damage to plant cells via oxidative stress involving the generation of ROS [31]. Diaminobenzidine (DAB) staining showed that H_2_O_2_ levels were largely increased in the leaves of *OX-GmbZIP15* soybean plants compared with WT plants as indicated by the larger amount of reddish-brown precipitate observed following the treatment with 200 mM NaCl and 300 mM mannitol (Figure 1B and Figure 2B). By contrast, the H_2_O_2_ contents in *35S:GmbZIP15-SRDX* soybean plants were comparable with that in WT plants under salt- or drought-stress conditions (Figure 1B and Figure 2B). We further investigated whether altered H_2_O_2_ contents reflected altered ROS-scavenging capability in these plants. For this, the activities of the two main antioxidant enzymes involved in ROS scavenging, namely, peroxidase (POD) and catalase (CAT), were determined in WT, *OX-GmbZIP15*, and *35S:GmbZIP15-SRDX* soybean seedlings before and after 24-h treatment with 200 mM NaCl and 300 mM mannitol. Before stress treatment, *OX-GmbZIP15* soybean plant showed higher activities of POD and CAT than WT plants (Appendix A), whereas following salt- and drought-stress treatments, there was a marked decrease in POD and CAT activities in *OX-GmbZIP15* soybean plant compared to WT plants (Appendix A). These results indicate that ROS scavenging was depressed in *GmbZIP15*-overexpressing soybean plants upon abiotic stress.

### 2.4. Changes of Stomatal Aperture in GmbZIP15 Transgenic Soybean Plants during Abiotic-Stress Conditions

Abiotic stress usually leads to a reduction in plant water loss through the regulation of the stomata aperture [32]. Thus, we analyzed stomatal regulation and its possible association with the stress-sensitive phenotype of *GmbZIP15* transgenic soybean plants. The stomatal apertures (width/length, W/L) of WT, *OX-GmbZIP15,* and *35S:GmbZIP15-SRDX* soybean plants were measured under control and abiotic-stress conditions. As shown in Figure 3, WT and transgenic soybean plants displayed similar stomatal apertures under control conditions. Moreover, following 200 mM NaCl and 300 mM mannitol treatments, there were no obvious differences in stomatal apertures between *35S:GmbZIP15-SRDX* and WT plants; however, a greater stomatal aperture was observed in *OX-GmbZIP15* plants compared to WT and *35S:GmbZIP15-SRDX* plants (Figure 3A,B). These data show that there is defective stomatal closure in *OX-GmbZIP15* plants upon abiotic stress.

### 2.5. Conservation of GmbZIP15-Mediated Abiotic-Stress Responses in Soybean and Arabidopsis

To further investigate the function of GmbZIP15 in response to abiotic stress, *OX-GmbZIP15* and *35S:GmbZIP15-SRDX* transgenic *Arabidopsis* plants were generated and two lines of each with higher expression levels were selected for further research (Figure 4B). Five-week-old soil-grown WT, *OX-GmbZIP15* and *35S:GmbZIP15-SRDX* transgenic *Arabidopsis* plant were watered with 150 mM NaCl. After 2 weeks, approximately 80% of WT (*n* = 50) and *35S:GmbZIP15-SRDX* plants (*n* = 50) remained viable, while nearly 90% of *OX-GmbZIP15* plants (*n* = 50) died (Appendix A). These results indicate that, similar to *OX-GmbZIP15* soybean plants, *OX-GmbZIP15 Arabidopsis* plants were also sensitive to salt stress. Similarly, following a 2-week dehydration treatment of WT, *OX-GmbZIP15*, and *35S:GmbZIP15-SRDX* transgenic *Arabidopsis* plants, approximately 85% of WT (*n* = 50) and *35S:GmbZIP15-SRDX* (*n* = 50) plants showed some degree of wilting phenotype but remained alive, whereas approximately 90% of the *OX-GmbZIP15* plants (*n* = 50) displayed a severe, near lethal wilting phenotype (Appendix A). These results suggest that *OX-GmbZIP15* overexpression causes similar drought- and salt-stress sensitivity in both *Arabidopsis* and soybean.

In addition, seed germination efficiency of WT, *OX-GmbZIP15,* and *35S:GmbZIP15-SRDX Arabidopsis* lines were evaluated under control and drought- and salt-stress conditions. For this, seeds of each line were germinated on 1/2 Murashige and Skoog Medium (MS) with or without 150 mM NaCl or 300 mM mannitol; the growth of two *OX-GmbZIP15* lines were inhibited severely (Figure 4A). In addition, the cotyledon greening rate was much lower in *OX-GmbZIP15* compared to WT and *35S:GmbZIP15-SRDX* transgenic *Arabidopsis* plants (Figure 4C). Taken together, these results indicate that GmbZIP15 plays a conserved role in drought- and salt-stress responses in both soybean and *Arabidopsis*.

To further understand the causal factors behind the drought- and salt-stress hypersensitivity of GmbZIP15-overexpressing *Arabidopsis* plants, we assayed expression levels of several known abiotic-stress-responsive genes in WT and *OX-GmbZIP15* transgenic *Arabidopsis* plants under control and drought- and salt-stress conditions. Transcript levels of each of the analyzed genes, including *AtWRKY33*, *AtCOR6-6*, *AtDREB2A,* and *AtRD29A*, were suppressed in *OX-GmbZIP15* plants compared with WT plants under normal conditions (Figure 4D,E). Under salt stress, the expression levels of *AtCOR6-6*, *AtDREB2A*, and *AtRD29A* were increased in both WT and *OX-GmbZIP15* plants following salt-stress treatment, although the magnitude of expression induction was much lower in *OX-GmbZIP15* plants (Figure 4D). Similar patterns of repressed expression of abiotic stress-responsive genes were detected in *OX-GmbZIP15* plants following drought-stress treatment. The expression levels of *AtWRKY33*, *AtDREB2A,* and *AtRD29A* were induced in both WT and *OX-GmbZIP15* plants by drought stress, but to a smaller extent in *OX-GmbZIP15* plants (Figure 4E). These results indicate that the drought- and salt-stress hypersensitivity caused by GmbZIP15 overexpression in *Arabidopsis* may be due to the repressed expression of drought- and salt-responsive genes.

*AtbZIP60* is the homologue of *GmbZIP15* in *Arabidopsis*. To investigate the role of AtbZIP60 in response to drought and salt stress, we analyzed the growth of the *bzip60* mutant (SALK_050203C) under control and drought- and salt-stress conditions. The results showed that the growth of the *bzip60* mutant was significantly repressed as compared to that in WT plants after 150 mM NaCl or 300 mM mannitol treatment (Figure 4A), which was accompanied by a lower cotyledon greening rate (Figure 4C), suggesting that these mutants are sensitive to drought and salt stress. These results agree with the previous findings that overexpression of *AtbZIP60* enhances salt, drought, and cold tolerance in rice [33]. Moreover, these data suggest that the roles of *GmbZIP15* and *AtbZIP60* in response to abiotic stress have diversified.

### 2.6. Transcriptomic Analysis of OX-GmbZIP15 Transgenic Soybean Plants in Response to Salt and Drought Stress

To further reveal the molecular mechanism behind the abiotic-stress sensitivity caused by *GmbZIP15* overexpression, we conducted RNA sequencing (RNA-seq) analysis using *OX-GmbZIP15-16* (OE) and WT soybean plants grown under either control conditions (mock treated) or treated with NaCl or mannitol. Three biological replicates were collected for each sample. In OE and WT plants, 2229 and 1693 differentially expressed genes (DEGs) were detected respectively, under salt-stress conditions compared to control conditions (fold change: ≥ 2 and *p* ≥ 0.05) (Figure 5A). Moreover, 8917 and 4811 DEGs were detected in OE and WT plants, respectively, under drought-stress conditions compared to control conditions (Figure 5A, Appendix A). Thus, there were more DEGs induced by both salt and drought stress in OE plants than in WT plants, indicating that *GmbZIP15* is responsible for gene expression changes upon salt and drought stress. In addition, we detected 1361 common DEGs (695 upregulated and 546 downregulated) in OE plants upon salt and drought stress (Figure 5B). In order to characterize the DEGs downregulated in the OE line upon salt and drought stresses, we studied their gene annotation (GO) term enrichment compared to that for untreated OE plants. As shown in Figure 5, a number of metabolic processes were enriched in both salt-stress and drought-stress downregulated gene sets in OE plants, such as response to stimuli, oxidation-reduction reactions, photosynthesis, hydrolase activity, phenylalanine biosynthesis, and some secondary metabolism processes (Figure 5C–F). These results imply that the above metabolic pathways are repressed in GmbZIP15-overexpressing soybean plants under abiotic stress.

### 2.7. GmbZIP15 Regulates the Expression of GmSAHH1, GmABF1, and GmWRKY12 in Soybean in Response to Abiotic Stress

On the basis of the RNA-seq data, five genes (FPKM > 100) with higher expression levels in OE plants compared with WT plants under normal conditions were selected for further analysis. However, only three of these genes (*GmSAHH1*, *GmWRKY12,* and *GmABF1*) were cloned successfully. *GmSAHH1* encodes a phosphate dehydrogenase, and expression of its *Arabidopsis* homologue (*ATSAHH1*; At4g13940) is detected in developing seeds and some anthers [34]. Moreover, the abundance of ATSAHH1 is reduced in protein extracts from salt-treated cells [35].

We performed a further qRT-PCR analysis to validate the RNA-seq data. Results consistently showed that the expression of *GmSAHH1* was higher in OE than in WT soybean plants under normal conditions, but lower in OE than in WT plants under salt- and drought-stress conditions (Appendix A). This repressed expression of *GmSAHH1* under abiotic stress was similar to that of *GmbZIP15*. To further investigate the biological function of *GmSAHH1* expression changes under abiotic stress, *GmSAHH1*-overexpressing (*OX-GmSAHH1*) transgenic *Arabidopsis* plants were obtained, and two lines with higher expression levels were selected for further research (Figure 6A) and then subjected to salt- and drought-stress treatments. Before treatment, no obvious morphological differences between 5-week-old WT and *OX-GmSAHH1* plants were observed. By contrast, under salt and drought stress, *OX-GmSAHH1* transgenic plants exhibited much more pronounced wilting compared with WT plants, which was almost lethal (Appendix A). Seedling growth was also significantly inhibited in *OX-GmSAHH1* plants upon salt and drought stress (Figure 6B). The similar salt- and drought-stress hypersensitivity of both *GmbZIP15-* and *GmSAHH1*-overexpressing plants suggests that GmbZIP15-regulated responses to salt and drought stress are likely mediated by *GmSAHH1* expression activation.

Previous work found that *GmWRKY12* positively regulates drought- and salt-stress responses in association with ABA and salicylic acid (SA), and *GmWRKY12* overexpression in soybean roots enhances soybean salt and drought tolerance [36]. In our RNA-seq data, *GmWRKY12* exhibited higher expression in OE plants than in WT plants under normal conditions and displayed increased expression in OE plants under stress conditions than normal conditions. We conducted a further qRT-PCR analysis and found that the expression of *GmWRKY12* was induced by salt and drought treatment in WT and OE plants (Appendix A). Moreover, we cloned an ABA-responsive gene (*GmABF1*) based on RNA-seq data, and its homologue in *Arabidopsis* (*AtABF1*, AT1G49720) is an ABA-dependent TF that regulates the expression of downstream ABA-inducible genes to improve plant drought resistance [10,37]. Our result demonstrated that its expression is induced by salt and drought stress (Appendix A). Furthermore, GmWRKY12-overexpressing (*OX-GmWRKY12*) and GmABF1-overexpressing (*OX-GmABF1*) seedlings and plants showed improved salt- and drought-stress tolerance compared to WT plants (Figure 6B and Appendix A), which agreed with the induced expression of *GmWRKY12* and *GmABF1* under salt and drought stress. These results suggest that *GmbZIP15* regulates plant salt- and drought-stress responses partly through inhibiting the expression of *GmWRKY12* and *GmABF1* upon abiotic stress.

## 3. Discussion

Abiotic stress, including salt and drought stress, has a considerable impact on the quality and yield of agricultural products. The important oilseed crop, soybean, is threatened by diverse categories of abiotic stress. Previous studies have indicated that bZIP TFs play diverse roles in response to various biotic and abiotic stress factors in different crop species, such as rice, soybean, rape, cotton, and maize [38,39,40,41]. In this study, a group-K bZIP TF, namely, GmbZIP15, was identified in soybean, and its functions in response to abiotic-stress conditions were analyzed in detail.

*AtbZIP60*, the *Arabidopsis GmbZIP15* homologue, positively modulates plant responses to salt, cold, and abiotic-stress conditions [33]. Here, we found that GmbZIP15 acts as a negative regulator of abiotic-stress responses. The transcription of *GmbZIP15* was suppressed by salt and drought stress (Appendix A) and GmbZIP15-overexpressing soybean displayed hypersensitivity to salt and drought stress (Figure 1A and Figure 2A). Our study suggests that *GmbZIP15* function in abiotic-stress responses differs from that of *AtbZIP60*, which is possibly due to the functional divergence of soybean and *Arabidopsis* during long-term evolution.

To adapt to abiotic stress, especially salt and drought stresses, the plants derive several strategies, including ion regulation and compartmentalization, induction of antioxidant enzymes, plant hormones and regulatory genes [42,43,44]. For example, the novel soybean regulatory gene GmTIP2;3 could effectively improve the tolerance of yeast to drought stress [43,45]. In addition, when under abiotic-stress conditions, plant endogenous ABA accumulates rapidly and activates the expression of stress-responsive genes, causing many physiological responses [37,46]. It has been demonstrated that ABA plays key roles in maintaining seed dormancy, inhibiting germination, and preventing seedling growth [47], and that abiotic stress is able to induce ABA biosynthesis and trigger ABA-dependent signaling pathways [48]. GmbZIP15 negatively regulates the expression of *GmABF1,* and the overexpression of *GmABF1* in *Arabidopsis* confers increased resistance to salt- and drought-stress conditions (Figure 6B). In addition, *GmWRKY12* in association with ABA positively regulates drought- and salt-stress responses, and the overexpression of *GmWRKY12* in soybean roots enhances plant salt and drought tolerance [36]. These results indicate a negative regulation of GmbZIP15 for ABA signaling which might via *GmABF1* and *GmWRKY12* in response to abiotic stress. In addition, the stress-responsive genes *AtDREB2A* and *AtRD29A* in *OX-GmbZIP15 Arabidopsis* plants (Figure 4C,D) and *GmDREBb* (Appendix A) in *OX-GmbZIP15* soybean plants exhibited lower transcript levels than those in WT plants under salt or drought conditions. AREBs and DREB are two groups of TFs that independently regulate the expression of genes involved in ABA-dependent and ABA-independent pathways [11]. The promoter regions of RD29 genes (RD29A and RD29B in *Arabidopsis*) are targeted by AREBs and DREBs; these genes encode hydrophilic proteins that endow plants with enhanced resistance to abiotic and cold stress [49]. Therefore, our study suggests that GmbZIP15 might act a negative regulator of plant drought- and salt-stress responses through ABA-dependent and ABA-independent pathways.

Abiotic stress potentially impairs plant cellular physiology and biochemistry via the excess generation of ROS [7,50,51]. For example, SlWRKY81 improved drought tolerance in tomato plants via the repression of SlP5CS1 transcription and thus reducing proline biosynthesis [52]. In this study, H_2_O_2_ contents sharply increased in *OX-GmbZIP15* plants compared to WT plants under drought- and salt-stress conditions (Figure 1B and Figure 2B). To control the level of ROS accumulation under stress conditions, plants have evolved a number of antioxidants, such as SOD, POD, and CAT, to scavenge ROS and to restore cellular redox homeostasis [53,54,55,56,57]. With the development of molecular biology, our understanding of molecular and physiology mechanisms is becoming clearer. Our results showed that the activities of POD and CAT were suppressed in *OX-GmbZIP15* transgenic soybean plants under salt and drought stress (Appendix A), indicating compromised ROS scavenging capability in *OX-GmbZIP15* plants in comparison with WT plants. Therefore, we hypothesize that GmbZIP15 plays a negative role in regulating these ROS-scavenging enzyme systems under abiotic stress.

Previous studies showed that bZIP TFs function in many biotic and abiotic-stress responses in plants through regulating diverse biochemical and physiological pathways [12,23,39,58]. RNA- sequencing has been widely used to investigate the molecular processes related to adaptive responses to abiotic stresses and to identify stress-resistance candidate genes by analyzing differences in transcript abundance [44]. In our research, we found that the functional annotation of DEGs that were enriched in the set of downregulated genes in OE plants after salt and drought treatment compared to that in control conditions indicated that, under salt or drought-stress conditions, GmbZIP15-regulated genes were mainly involved in processes such as oxidoreductase activity, phenylalanine biosynthesis, phosphotransferase activity, and some secondary metabolism (Figure 5). As an important polyphenolic secondary metabolite, isoflavones play a crucial role in plants facing diverse environmental-stress conditions [59,60,61]. *PtSAP13*, for example, enhances salt tolerance by upregulating the transcript level of stress-responsive genes and inducing multiple biological pathways, such as phenylalanine biosynthesis and dioxygenase activity [62], thus implying that the phenylalanine metabolism process is involved in GmbZIP15-regulated abiotic-stress responses (Figure 5C,E). In addition, photosynthesis is essential for plant growth and is important for plants to maintain a balance between growth and stress responses [63,64]. For example, when cyanobacteria grow under stress conditions, photosynthesis-related genes are usually downregulated, whereas stress response-related genes are upregulated [65]. Water deficiency significantly affects photosynthetic characteristics. The drought-tolerant soybean cultivar displayed the maximum values of chlorophyll fluorescence (Fv/Fm, qP, ϕPSII, and ETR) [66]. When under environmental stress, plants will close their stomata and thus restrict the entry of CO_2_ into the leaf and reduce the rate of photosynthesis [67]. In this study, many downregulated genes in *OX-GmbZIP15* transgenic plants after drought and salt stress were associated with photosynthesis (Figure 5C–F), suggesting that stress adaption was prioritized over photosynthesis; however, impaired stomatal aperture regulation in *OX-GmbZIP15* affected plant survival. In addition, enzyme-catalyzed removal of ROS such as superoxide and H_2_O_2_ are important in plant survival under stress conditions [54,63]. As observed here, those genes downregulated in response to abiotic stress represented antioxidant-related processes. Taken together, multiple metabolic pathways seem to be involved in the GmbZIP15-mediated abiotic-stress response network.

In summary, overexpression of *GmbZIP15* in soybean resulted in hypersensitivity to salt and drought stresses compared with wild-type (WT) plants, which was associated with lower transcript levels of stress-responsive genes, defective stomatal aperture regulation, and reduced antioxidant enzyme activities. Furthermore, RNA-seq and qRT-PCR analyses revealed that *GmbZIP15* positively regulates *GmSAHH1* expression and negatively regulates *GmWRKY12* and *GmABF1* expression in response to salt and drought stresses (Figure 7). These data provided new information for understanding the function of *GmbZIP15* and might facilitate the improvement of plant abiotic-stress tolerance through genetic manipulation in the future.

## 4. Materials and Methods

### 4.1. Vector Construction and Transformation

To generate the *OX-GmbZIP15* construct, the *GmbZIP15* (Glyma.02G161100) coding DNA sequence (CDS) was amplified and the PCR fragments were cloned into the pENTR/D-TOPO vector (Invitrogen, Carlsbad, CA, USA). The pENTR clones were recombined into the destination vector pGWB506 using LR Clonase II (Invitrogen). The resulting construct also contained the selectable marker BAR for glufosinate resistance [68].

*35S: GmbZIP15-SRDX* was generated by amplifying GmbZIP15 cDNA sequence and an SRDX motif was added to the end of the cDNA sequence (ctagatctggatctagaactccgtttgggtttcgcttaa). The PCR fragment was cloned into the pENTR/D-TOPO vector (Invitrogen), and the pENTR/D-TOPO clones were recombined into the destination vector pGWB506 using LR Clonase II (Invitrogen) [69]. The vectors *OX-GmbZIP15* and *35S: GmbZIP15-SRDX* were then transformed into soybean by agrobacterium-mediated method [70] and the soybean genotype C03-3 was used.

*GmWRKY12* (Glyma.01G224800)-, *GmABF1* (GmbZIP157, Glyma.20G049200)-, and *GmSAHH1* (Glyma.08G108800)-overexpressing vectors were constructed as above [68]. WT *Arabidopsis* (Col-0) plants were then infected with the transformed bacteria by the floral dip method [71]. All the primers used in the article were listed in Appendix A.

### 4.2. Plant Materials and Stress Treatments

Soybean plant seeds including WT (C03-3) and transgenic *GmbZIP15* plants were grown for 15 days in pots containing nutritional soil and vermiculite in green house. The seedlings were then exposed to drought and salt stresses. For drought stress, the soybean seedlings were watered with 300 mM mannitol to induce the rapid drought stress. For salt treatment, the seedlings were transferred to 200 mM NaCl solution. All seedlings leaves were harvested at 0, 6, 12, and 24 h under stress conditions for RNA extraction.

*Arabidopsis* ecotypes Col-0 was used in this study. The T-DNA mutant *Atbzip60* (SALK_050203C) was obtained from the *Arabidopsis* Biological Resource Centre (ABRC). All Seeds were germinated on 1/2 MS medium containing NaCl or mannitol, after vernalization at 4 °C for 3 days, the plates containing the seeds were placed in a growth chamber with temperature 22 °C, and a photoperiod of 16 h light/8 h dark.

### 4.3. Diaminobenzidine (DAB) Staining

Following previously described methods for hydrogen peroxide (H_2_O_2_) detection [72], the soybean leaves after salt and drought treatment for 4 days were immediately vacuum-infiltrated for 20 min with Tris-HCl (pH 7.4) containing 1% (*w/v*) DAB. Thereafter, all the leaves were placed in light for 10 h then boiled for 20 min in 75% ethanol.

### 4.4. Determination of Stomatal Aperture

The fully expanded leaves of 2-week-old soybean plants were floated in the stomatal opening buffer with 30 mM KCl and 10 mM MES-KOH, pH 6.15 for 2 h under a cool white light, and then 200 mM NaCl or 300 mM mannitol were added to the opening buffer [22]. After 1 h, the subepidermal peels were stripped and used for stomatal aperture measurements under the microscope. In addition, different phytohormones were added to the opening buffer and the stomatal apertures were observed at different timepoints.

### 4.5. RNA Extraction and Quantitative qRT-PCR

Total RNA was extracted using Trizol (Invitrogen, Carlsbad, CA, USA) then reverse-transcribed using the PrimeScript RT-PCR kit (TaKaRa) [71]. The relative expression levels of selected genes were detected by qRT-PCR using Bio-Rad QRT-PCR system (Foster City, CA, USA) and SYBR Premix Ex Taq II (TaKaRa Perfect Real Time). The qRT-PCR program was 95 °C for 30 s; 40 cycles of 95 °C for 5 s and 60 °C for 34 s; and 95 °C for 15 s [68]. GmActin was used for normalization.

### 4.6. Determination of Antioxidant Enzyme Activity

The antioxidants including peroxidase (POD) and catalase (CAT) were extracted from approximately 0.1 g of soybean leaves using 1 mL extraction solution. The 2-week-old seedlings were treated with 200 mM NaCl or 300 mM mannitol for 24 h and then the leaves were harvested. The enzyme activities were measured according to the protocol from Solarbio Biochemical Assay Division.

### 4.7. RNA-Seq Data Analysis

Leaves of 2-week-old soybean plants including WT and *OX-GmbZIP15-16* plants treated with 200 mM NaCl or 300 mM mannitol were harvested at 24 h for RNA-seq, and three biological replicates were analyzed. The libraries were constructed by BGI (Beijing Genomics Institute) then sequenced. GO analyses were performed using the agriGO online toolkit [http://bioinfo.cau.edu.cn/agriGO/index.php].

## Figures and Tables

**Figure 1 ijms-21-07778-f001:**
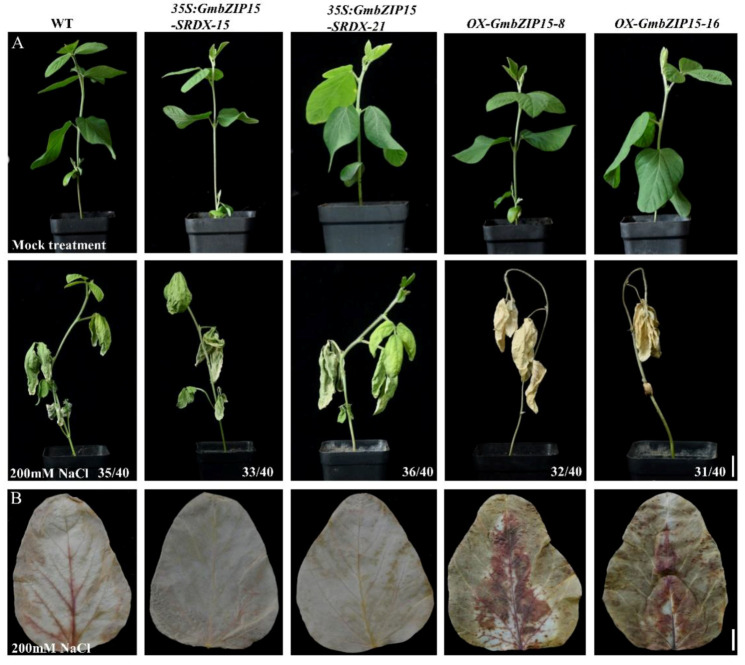
*GmbZIP15* negatively regulates salt-stress resistance in soybean. (**A**) Phenotype observation of transgenic soybean seedlings in response to salt stress. The pictures were obtained before or after 200 mM NaCl treatment for 2 weeks. Numbers in the panels denote the frequencies of the phenotypes shown. (**B**) Diaminobenzidine (DAB) staining of the soybean leaves. All the plants were treated with 200 mM NaCl for 4 days and then the leaves were harvested. The depth of color shows the H_2_O_2_ content in leaves. Bar = 1 cm.

**Figure 2 ijms-21-07778-f002:**
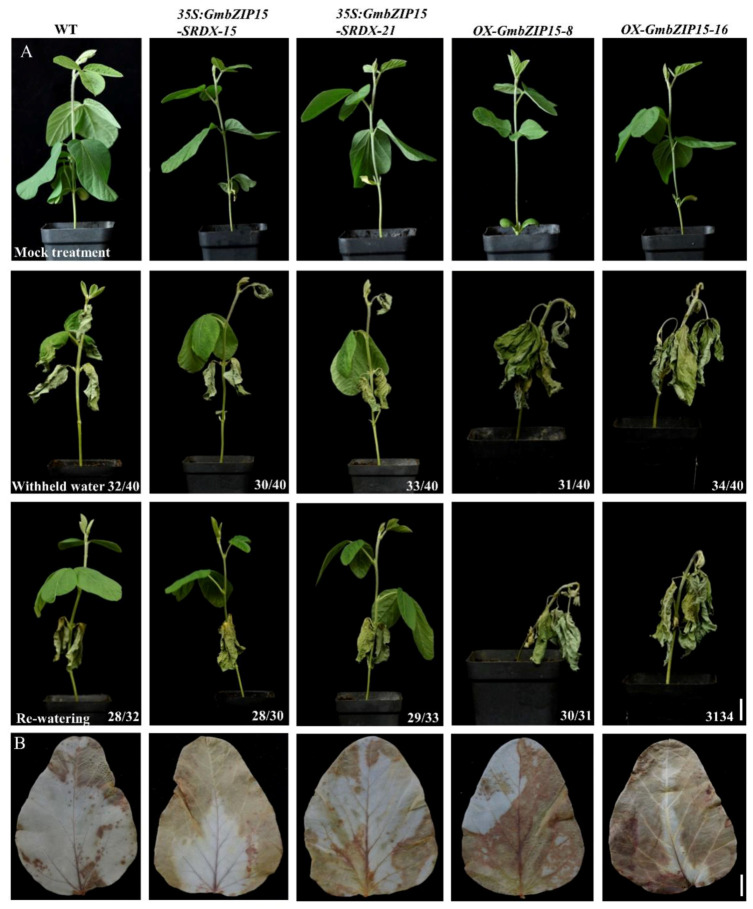
*GmbZIP15* negatively regulates drought-stress resistance in soybean. (**A**) Phenotype observation of transgenic soybean seedlings in response to drought stress. The pictures were obtained under normal conditions; thereafter, the plants were not watered for 2 weeks, then rewatered for 3 days. Numbers in the panels denote the frequencies of the phenotypes shown. (**B**) DAB staining of the soybean leaves. All the plants were not watered for 4 days and then the leaves were harvested. The depth of color shows the H_2_O_2_ content in the leaves. Bar = 1 cm.

**Figure 3 ijms-21-07778-f003:**
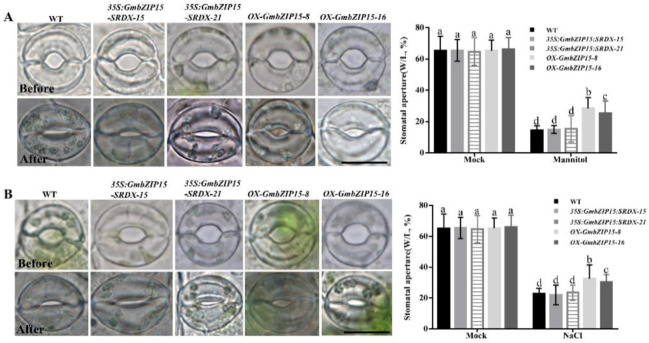
Changes in the stomatal aperture in *GmbZIP15* transgenic soybean plants under salt- and drought-stress conditions. (**A**,**B**) Comparison of stomatal aperture with width over length before or after 200 mM NaCl treatment for 1 h (**A**) or before or after 300 mM mannitol treatment for 1 h (**B**). Data were calculated from 100 stomata of the leaves of three different soybean plants. The experiments were performed three times with similar results. Bar = 10 μm. Errors bars indicate ± SD of three biological replicates. Significant differences between samples labeled a, b, and c were determined by one-way ANOVA, *p* < 0.05.

**Figure 4 ijms-21-07778-f004:**
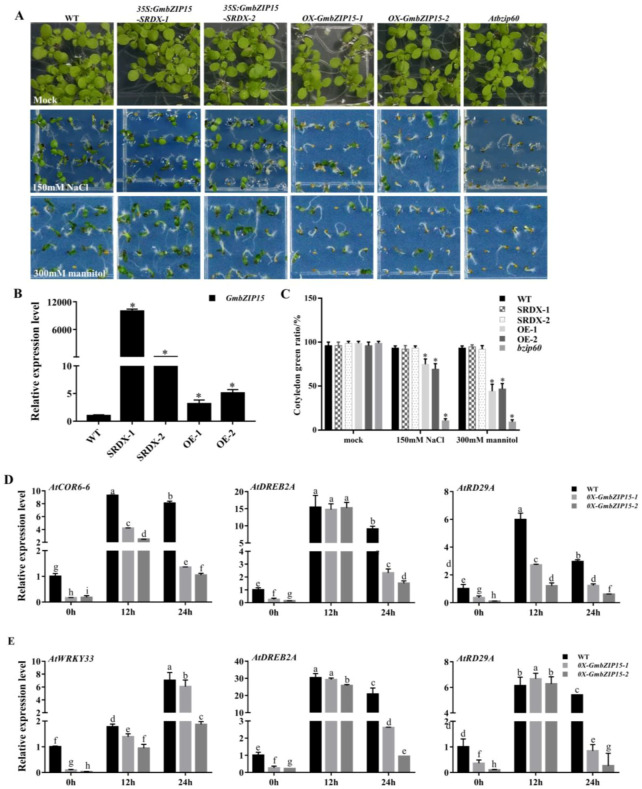
*GmbZIP15*-overexpressed *Arabidopsis* is hypersensitive to salt and drought stresses. (**A**) Phenotype observation of wild-type (WT) and *GmbZIP15* transgenic *Arabidopsis* plants under normal and stress conditions. All the seeds were germinated on the 1/2 Murashige and Skoog Medium (MS) medium under normal conditions or supplemented with 150 mM NaCl or 300 mM mannitol for 1 week. (**B**) Transcript level detection of *GmbZIP15* in transgenic *Arabidopsis* plants. (**C**) Quantification of the cotyledon green rate. (**D**,**E**) *GmbZIP15* regulates stress-responsive gene expression in WT and *GmbZIP15* transgenic *Arabidopsis* plants. Gene expression levels of *AtCOR6-6*, *AtDREB2A,* and *AtRD29A* were quantified by qRT-PCR assays after 150 mM NaCl treatment for 0.12, and 24 h (**D**). Gene expression levels of *AtWRKY33*, *AtDREB2A,* and *AtRD29A* were quantified by qRT-PCR assays after 300 mM mannitol treatment for 0.12, and 24 h (**E**). Errors bars indicate ± SD of three biological replicates. Significant differences between samples labeled a, b, and c were determined by one-way ANOVA, *p* < 0.05.

**Figure 5 ijms-21-07778-f005:**
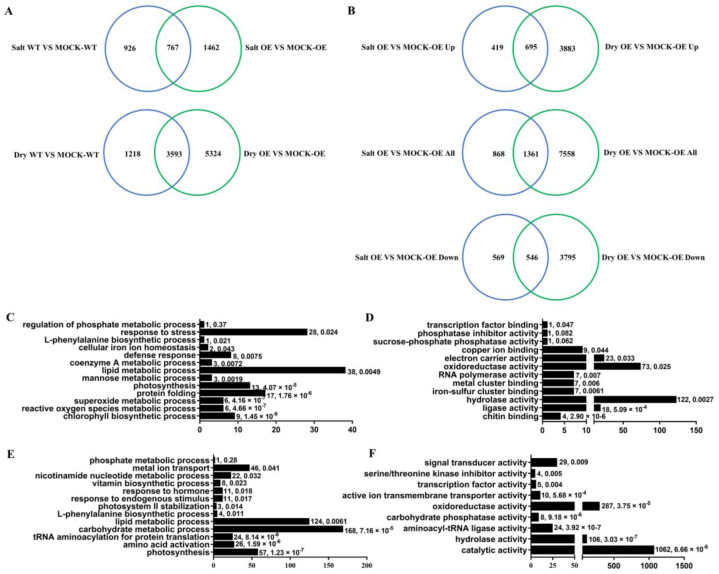
Transcriptomic analysis of *OX-GmbZIP15* transgenic soybean plants. (**A**) Number of specific and common salt- and drought-responsive differentially expressed genes (DEGs) in the WT and *OX-GmbZIP15-16* soybean plants. (**B**) Number of specific and common DEGs in the *OX-GmbZIP15-16* soybean plants after salt and drought-stress treatment. (**C**,**D**) gene annotation (GO) analysis of the DEGs downregulated in *OX-GmbZIP15-16* soybean plants after salt stress: (**C**) biological process; (**D**) molecular function. (**E**,**F**) GO analysis of the DEGs downregulated in *OX-GmbZIP15-16* soybean plants after drought stress; (**E**) biological process; (**F**) molecular function. The numbers next to the columns indicate the number of DEGs with corresponding annotation and the *p*-value, respectively (**C**–**F**).

**Figure 6 ijms-21-07778-f006:**
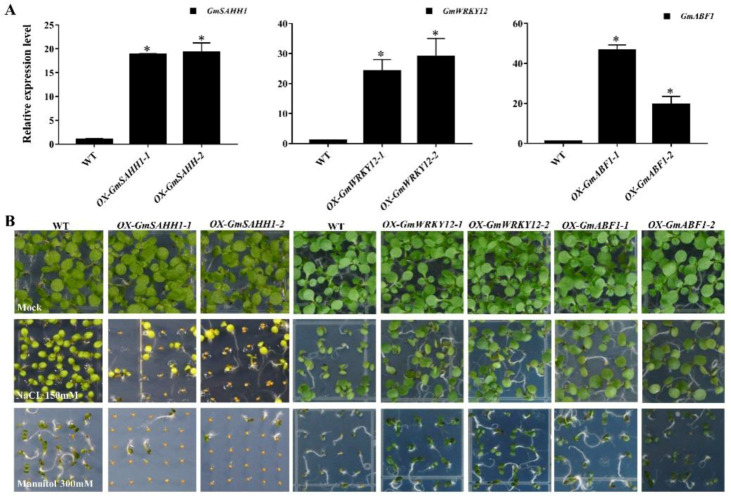
Phenotypic analysis of *GmSAHH1-*, *GmWRKY12-,* and *GmABF1*-overexpressed *Arabidopsis* plants in response to salt and drought stresses. (**A**) Transcript level detection of *GmSAHH1*, *GmWRKY12,* and *GmABF1* in transgenic *Arabidopsis* plants. Errors bars indicate ± SD of three biological replicates. Significant differences between samples labeled asterisks were determined by one-way ANOVA, *p* < 0.05. (**B**) Growth observation of WT and overexpression of *GmSAHH1*, *GmWRKY12,* and *GmABF1 Arabidopsis* seedlings under either normal conditions or 150-mM-NaCl- and 300-mM-mannitol-supplemented 1/2 MS medium.

**Figure 7 ijms-21-07778-f007:**
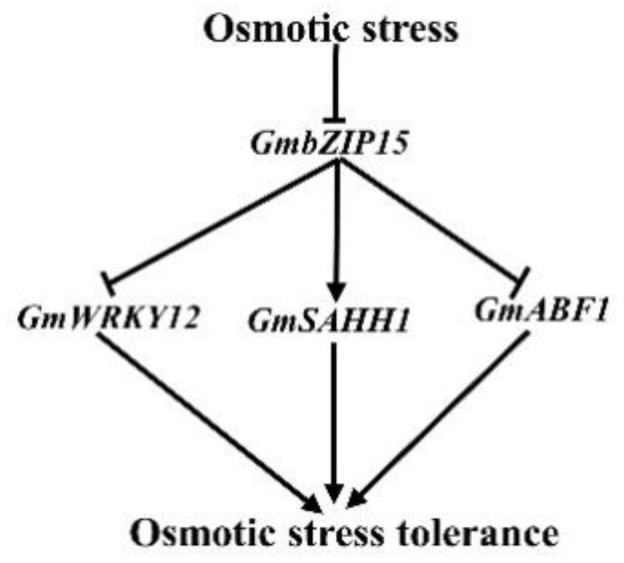
A schematic model of *GmbZIP15* mediated abiotic-stress tolerance in soybean. *GmbZIP15* negatively modulates the abiotic-stress tolerance: *GmbZIP15* positively regulates the expression of *GmSAHH1* and negatively regulates the expression of *GmWRKY12* and *GmABF1* in response to abiotic stresses. The arrows indicate induction or positive modulation; the blunt-end arrows represent block or suppression.

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
