# Peer review of "The bZIP Transcription Factor GmbZIP15 Negatively Regulates Salt- and Drought-Stress Responses in Soybean"

_ijms, 2020, doi:10.3390/ijms21207778_

Round 1

Reviewer 1 Report

The topic of the research is coherent to the aims of IJMS. The paper is well introduced.  The title reflects the complexity of the paper and the results are discussed convincingly. The literary structure of the introduction is good, containing key information about the work and on the problem under study. The manuscript presents some new findings and has high novelty, and particularly, the results of the role of different plant stress mechanisms. 

 The authors characterized the transcript level of GmbZIP15  and overexpression of GmbZIP15 in soybean resulted in hypersensitivity to osmotic stress, overexpression of GmbZIP15 in soybean resulted in hypersensitivity. The data indicate that GmbZIP15 functions as a negative regulator in response to salt and drought stresses.

Materials and methods are relatively well presented. 

The paper is clear and well-written. The style is adequate.  The manuscript presents interesting research.

Results reported have not been published elsewhere.

Experiments, statistics, and other analyses are performed to a high technical standard and are described in sufficient detail. Discussion and the writing should be improved to make the manuscript easy to follow,  which made this work stronger and more attractive. Add more pieces of information about the molecular and physiological flexibility of plants under drought and salt stress. The discussion section needs to revise.

Specific comments:

- standard parameters were determined. Write more exactly, which kind of different phytohormones were added to the opening buffer and the stomatal apertures?

- missing critical discussion and a deeper analysis of photosynthetic parameters and processes of osmotic adjustment

- the discussion should be improved to make the manuscript easy to follow, and more attractive.

- I would like to advise to add new pieces of information about biochemical and molecular mechanisms.

- arguments need clearer and tighter presentation. The understanding of physiological mechanisms is limited, as it is restricted to papers that have a particular view and deliberately ignore alternatives, and does not present a balanced view of the evidence

- conclusions are presented in an appropriate fashion and are supported by the data, however, authors could add new concrete conclusions, future perspectives. That is,  the body of the text has to be improved in the discussion

Please  read/use the following references:

Ahammed GJ, Li X, Wan H, Zhou G, Cheng Y. SlWRKY81 reduces drought tolerance by attenuating proline biosynthesis in tomato. Scientia Horticulturae. 2020;270. doi: 10.1016/j.scienta.2020.109444

Iqbal N, Hussain S, et al.: Drought Tolerance of Soybean (Glycine max L. Merr.) by Improved Photosynthetic Characteristics and an Efficient Antioxidant Enzyme Activities Under a Split-Root System. Front. Physiol. 2019, 10:786. doi: 10.3389/fphys.2019.00786

Guo, H., Zhang, L., Cui, Y.N., Wang, S.M., and Bao, A.K., 2019. Identification of candidate genes related to salt tolerance of the secretohalophyte Atriplex canescens by transcriptomic analysis. BMC Plant Biol. 19

Tang X., Mu X., Shao H., et al. (2015): Global plant-responding mechanisms to salt stress: physiological and molecular levels and implications in biotechnology, Critical Reviews in Biotechnology, 35:4, 425-437, DOI: 10.3109/07388551.2014.889080

Agarwal, P. K., Shukla, P. S., Gupta, K., and Jha, B. (2013). Bioengineering for salinity tolerance in plants: state of the art. Mol. Biotechnol. 54, 102-123. doi: 10.1007/s12033-012-9538-3

Zhang D, Tong J, He X, et al.: A Novel Soybean Intrinsic Protein Gene, GmTIP2;3, Involved in Responding to Osmotic Stress. Front. Plant Sci. 2016, 6:1237. doi: 10.3389/fpls.2015.01237

Overall, the MS may be accepted for publication in IJMS after minor revisions as mentioned above

Author Response

Reviewer 1

The topic of the research is coherent to the aims of IJMS. The paper is well introduced.  The title reflects the complexity of the paper and the results are discussed convincingly. The literary structure of the introduction is good, containing key information about the work and on the problem under study. The manuscript presents some new findings and has high novelty, and particularly, the results of the role of different plant stress mechanisms. 

 The authors characterized the transcript level of GmbZIP15  and overexpression of GmbZIP15 in soybean resulted in hypersensitivity to osmotic stress, overexpression of GmbZIP15 in soybean resulted in hypersensitivity. The data indicate that GmbZIP15 functions as a negative regulator in response to salt and drought stresses.

Materials and methods are relatively well presented. 

The paper is clear and well-written. The style is adequate.  The manuscript presents interesting research.

Results reported have not been published elsewhere.

Experiments, statistics, and other analyses are performed to a high technical standard and are described in sufficient detail.

Discussion and the writing should be improved to make the manuscript easy to follow,  which made this work stronger and more attractive. Add more pieces of information about the molecular and physiological flexibility of plants under drought and salt stress. The discussion section needs to revise.

Response: we added more pieces of information about the molecular and physiological flexibility of plants under drought and salt stress.

Specific comments:

- standard parameters were determined. Write more exactly, which kind of different phytohormones were added to the opening buffer and the stomatal apertures?

Response: 200mM NaCl or 300Mm mannitol were added to the opening buffer, and then soybean plants leaves were treated for 1 hour, this was added in line 424-425.

- missing critical discussion and a deeper analysis of photosynthetic parameters and processes of osmotic adjustment.

Response: we added a deeper analysis of photosynthetic and processes of osmotic adjustment in discussion part line 377-379.

- the discussion should be improved to make the manuscript easy to follow, and more attractive.

Response: we had added some new sight evidence in the discussion part.

- I would like to advise to add new pieces of information about biochemical and molecular mechanisms.

Response: we had added new pieces of information about biochemical and molecular mechanisms in the discussion part in line 324-327.

- arguments need clearer and tighter presentation. The understanding of physiological mechanisms is limited, as it is restricted to papers that have a particular view and deliberately ignore alternatives, and does not present a balanced view of the evidence.

Response: we had added new pieces of information about physiological mechanisms in the discussion part in line 348-349, 361-363.

- conclusions are presented in an appropriate fashion and are supported by the data, however, authors could add new concrete conclusions, future perspectives. That is, the body of the text has to be improved in the discussion.

Response: we added new concrete conclusions, future perspectives in the conclusion part.

Please read/use the following references:

Ahammed GJ, Li X, Wan H, Zhou G, Cheng Y. SlWRKY81 reduces drought tolerance by attenuating proline biosynthesis in tomato. Scientia Horticulturae. 2020;270. doi: 10.1016/j.scienta.2020.109444

Iqbal N, Hussain S, et al.: Drought Tolerance of Soybean (Glycine max L. Merr.) by Improved Photosynthetic Characteristics and an Efficient Antioxidant Enzyme Activities Under a Split-Root System. Front. Physiol. 2019, 10:786. doi: 10.3389/fphys.2019.00786

Guo, H., Zhang, L., Cui, Y.N., Wang, S.M., and Bao, A.K., 2019. Identification of candidate genes related to salt tolerance of the secretohalophyte Atriplex canescens by transcriptomic analysis. BMC Plant Biol. 19

Tang X., Mu X., Shao H., et al. (2015): Critical Reviews in Biotechnology, 35:4, 425-437, DOI: 10.3109/07388551.2014.889080

Agarwal, P. K., Shukla, P. S., Gupta, K., and Jha, B. (2013). Bioengineering for salinity tolerance in plants: state of the art. Mol. Biotechnol. 54, 102-123. doi: 10.1007/s12033-012-9538-3

Zhang D, Tong J, He X, et al.: A Novel Soybean Intrinsic Protein Gene, GmTIP2;3, Involved in Responding to Osmotic Stress. Front. Plant Sci. 2016, 6:1237. doi: 10.3389/fpls.2015.01237

Overall, the MS may be accepted for publication in IJMS after minor revisions as mentioned above

Reviewer 2 Report

I believe that the work presents some critical points in the applied methods,

The use of mannitol to determine osmotic stress raises me many perplexities.

Mannitol is synthesized by various plants in response to water stresses. This polymer can interact with the transport channels for ions and water in conditions of water stress.

In practice it is a biostimulant which interacts with the biochemistry of the plant determines tolerance to water stress.

It would have been more appropriate to use PEG 6000.

Did mannitol and NaCl treatments have the same osmotic pressure?

NaCl causes osmotic stress and toxicity

It is important that the osmotic pressures determined by the two treatments are identical, only in this case it is possible to calculate the additional toxicity effect determined by NaCl

I ask that the actual osmotic pressure has not been measured.

Author Response

Reviewer 2

I believe that the work presents some critical points in the applied methods,

The use of mannitol to determine osmotic stress raises me many perplexities.

Mannitol is synthesized by various plants in response to water stresses. This polymer can interact with the transport channels for ions and water in conditions of water stress.

In practice it is a biostimulant which interacts with the biochemistry of the plant determines tolerance to water stress.

It would have been more appropriate to use PEG 6000.

Response: because we also used 1/2 MS to grow the transgenic Arabidopsis seeds, when we added the PEG 6000 into the 1/2 MS, it can not be solid, so we can not use PEG 6000 to perform drought stress. We also read some researches focused on the abiotic stress published in 2020, we found that many people also used mannitol to performed drought stress, so we think it is also appropriate to use mannitol or PEG 6000?

  1. Luo, X.; Li, C.; He, X.; Zhang, X.; Zhu, L., ABA signaling is negatively regulated by GbWRKY1 through JAZ1 and ABI1 to affect salt and drought tolerance. Plant cell reports 2020, 39 (2), 181-194.
  2. Wang, N.; Liu, Y.; Cai, Y.; Tang, J.; Li, Y.; Gai, J., The soybean U-box gene GmPUB6 regulates drought tolerance in Arabidopsis thaliana. Plant Physiol Biochem 2020, 155, 284-296.
  3. Li, X.; Tang, Y.; Zhou, C.; Zhang, L.; Lv, J., A Wheat WRKY Transcription Factor TaWRKY46 Enhances Tolerance to Osmotic Stress in transgenic Arabidopsis Plants. Int J Mol Sci 2020, 21 (4).
  4. Ju, Y. L.; Yue, X. F.; Min, Z.; Wang, X. H.; Fang, Y. L.; Zhang, J. X., VvNAC17, a novel stress-responsive grapevine (Vitis vinifera L.) NAC transcription factor, increases sensitivity to abscisic acid and enhances salinity, freezing, and drought tolerance in transgenic Arabidopsis. Plant Physiol Biochem 2020, 146, 98-111.

 Did mannitol and NaCl treatments have the same osmotic pressure?

Response: in this research, we focused on the function of GmbZIP15 in response to salt and drought stress. After salt stress treatment for 6h, 12h, the expression of GmbZIP15 decreased more seriousness in overexpression plants than drought stress treatment, but till 24h, the expression of GmbZIP15 was similar.  

NaCl causes osmotic stress and toxicity

It is important that the osmotic pressures determined by the two treatments are identical, only in this case it is possible to calculate the additional toxicity effect determined by NaCl.

 Response: thank you for your suggestion, our research was aim to identify the function of GmbZIP15 in response to abiotic stresses, as salt and drought stress are two major abiotic stresses, so might be our consider is ambiguous, so we decided to change the description “osmotic stress” to “abiotic stress”.

I ask that the actual osmotic pressure has not been measured.

Round 2

Reviewer 2 Report

Most of my doubts about the exact type of tress studied have been clarified.